# DEBIAS THE TRAINING OF DIFFUSION MODELS

## ABSTRACT

Diffusion models have demonstrated compelling generation quality by optimizing the variational lower bound through a simple denoising score matching loss. In this paper, we provide theoretical evidence that the prevailing practice of using a constant loss weight strategy in diffusion models leads to biased estimation during the training phase. Simply optimizing the denoising network to predict Gaussian noise with constant weighting may hinder precise estimations of original images. To address the issue, we propose an elegant and effective weighting strategy grounded in the theoretically unbiased principle. Moreover, we conduct a comprehensive and systematic exploration to dissect the inherent bias problem deriving from constant weighting loss from the perspectives of its existence, impact and reasons. These analyses are expected to advance our understanding and demystify the inner workings of diffusion models. Through empirical evaluation, we demonstrate that our proposed debiased estimation method significantly enhances sample quality without the reliance on complex techniques, and exhibits improved efficiency compared to the baseline method both in training and sampling processes.

## 1 INTRODUCTION

Diffusion models (Sohl-Dickstein et al., 2015; Ho et al., 2020) have emerged as powerful generative models that garner significant attention recently. Their popularity stems from the remarkable ability to generate diverse and high-quality samples (Dhariwal & Nichol, 2021; Rombach et al., 2022; Ramesh et al., 2022; Nichol & Dhariwal, 2021) as well as the training-stable loss form, compared to the adversarial training paradigms used in Generative Adversarial Networks (GANs) (Goodfellow et al., 2014). Diffusion models often serve as a fundamental building block and have exhibited impressive success on a wide rang of tasks (Ruiz et al., 2023; Saharia et al., 2022). While, it is usually employed as a black-box component in these works. Some methods delve into the methodology of diffusion models, particularly focusing on the reverse sampling process. They typically aim to minimize the number of steps in the sampling process to accelerate generation, achieved through designing more efficient noise schedule (Song et al., 2020; Liu et al., 2021; Lu et al., 2022) or progressive distilling (Salimans & Ho, 2022; Meng et al., 2023). These works make commendable attempts to enhance fundamental diffusion models and provide insights into the mechanisms concealed within the black box.

Here we focus on the perspective of training diffusion models, which traditionally retain an elegantly simple loss function, i.e., the L2 loss with constant weight between the Gaussian noise and the predicted one as follows:

$$L = \sum_t \mathbb{E}_{x_0, \epsilon} \left[ ||\epsilon - \epsilon_\theta(x_t, t)||^2 \right]. \tag{1}$$

Several works have recognized the issue of using this constant weighting and proposed alternative weights and objectives (Choi et al., 2022; Salimans & Ho, 2022; Hang et al., 2023). Nonetheless, it is still an open question: whether such a constant weighting form is not optimal and, if so, why and how it affects the model's performance.

In this paper, we theoretically demonstrate the suboptimality of the constant-weighting loss formulation, revealing its potential to introduce biased estimations during training and diminish the model's performance. Consequently, we propose an effective and elegant debiased loss weight to address the issue, adhering to the debiased principle. Apart from the theoretical proof and the debiased

solution, more importantly, we figure out several key questions crucial for achieving a systematical understanding of the bias problem in conventional diffusion models, from its existence, impact, and reasons. First, we show the existence of the biased estimation problem in the training process. Concretely, the output of denoising network can be very close to the target Gaussian noise at every step $t$, while, the corresponding estimated $\hat{x}_0$ may severely deviates from $x_0$, which grows as $t$ gets larger. Then, we analyse the impact of the biased estimation problem on the sampling process, which we call *biased generation*. Biased generation mainly attributes to the chaos and inconsistency in the early few sampling steps, which affects the final generation via error propagation effect. And the final results of biased generation usually come with poor details, color shift and global inconsistency. Additionally, we unravel the underlying causes of biased estimation. The importance and optimization difficulty of the denoising network is vastly different at different step $t$.

We empirically show that the proposed debiased estimation method is capable of addressing the above problems and substantially elevates the sample quality. Our method shows enhanced efficiency in both the training and sampling, achieving superior performance to previous constant weighting strategy with much less training iterations and sampling steps. All these are achieved by slightly revising the loss weight strategy with only one additional line of code, which is orthogonal to existing sampling accelerating methods. We expect the in-depth analysis of dissecting the bias issue can provide a further understanding for the research of diffusion models.

## 2 BACKGROUND

Diffusion models (Sohl-Dickstein et al., 2015; Ho et al., 2020) transform complex data distribution $p_{data}(x)$ into simple noise distribution $\mathcal{N}(0, \mathbf{I})$ and learn to recover data from noise. It contains two processes: the *forward diffusion process* and the *reverse denoise process*.

The *forward diffusion process* starts from a clean data sample $x_0$ and repeatedly injects Gaussian noise according to the transition kernel $q(x_t|x_{t-1})$ as follows:

$$q(x_t|x_{t-1}) = N(x_t; \sqrt{1-\beta_t}x_{t-1}, \beta_t I), \tag{2}$$

where $\beta_t$ can be learned or held constant as hyper-parameters, controlling the variance of noise added at each step. For example, Ho et al. (Ho et al., 2020) employed a linear noise schedule, and Nichol et al. (Nichol & Dhariwal, 2021) applied a cosine schedule. From the Gaussian diffusion process, we can derive closed-form expressions for the marginal1 distribution $q(x_t|x_0)$ as follows:

$$x_t = \sqrt{\alpha_t}x_0 + \sqrt{1-\alpha_t}\epsilon, \tag{3}$$

where $\epsilon \sim \mathcal{N}(0, \mathbf{I})$ and $\alpha_t := \prod_{s=1}^{t}(1 - \beta_s)$. Note that the above-defined forward diffusion formulation has no learnable parameters, and the reverse diffusion step cannot be applied due to having no access to $x_0$ in the inference stage. Therefore, we further introduce the learnable reverse denoise process for estimating $x_0$ from $x_T$.

The *reverse denoise process* is trained to reverse the forward diffusion process in Eq. 2 by learning the denoise network, with the current de facto training objective being Eq. 1. Given a randomly sampled Gaussian noise $x_T \sim \mathcal{N}(0, \mathbf{I})$, the sample iteratively gets less noisy as follows:

$$x_{t-1} = \frac{1}{\sqrt{1-\beta_t}}(x_t - \frac{\beta_t}{\sqrt{1-\alpha_t}}\epsilon_\theta(x_t, t)) + \sigma_t z, \tag{4}$$

where $\sigma_t^2$ is a variance and $z \sim \mathcal{N}(0, \mathbf{I})$. Ho et al. (Ho et al., 2020) used $\beta_t$ as $\sigma_t^2$.

Kingma et al. (Kingma et al., 2021) proposed the use of *signal-to-noise ratio* (SNR) to represent the noise schedules in diffusion models. The SNR of the intermediate noisy sample $x_t$ is calculated as the ratio of the squared mean and variance, which is expressed as:

$$\text{SNR}(t) = \alpha_t/(1 - \alpha_t). \tag{5}$$

## 3 THEORETICAL EXPLORATION OF THE INHERENT BIAS

### 3.1 TRAINING OBJECTIVES OF CONVENTIONAL DIFFUSION MODELS

Diffusion models are trained by optimizing a variational lower bound (VLB). For each step $t$, the denoising score matching loss $L_t$ is the distance between two Gaussian distributions, which can be

rewritten as:

$$L_t = D_{KL}(q(x_{t-1}|x_t, x_0) \,||\, p_\theta(x_{t-1}|x_t)), \tag{6}$$

where the reverse diffusion step $q(x_{t-1}|x_t, x_0)$ and $p_\theta(x_{t-1}|x_t)$ can be expressed as follows:

$$
\begin{aligned}
q(x_{t-1}|x_t, x_0) &= N(x_{t-1}; \tilde{\mu}_t(x_t, x_0), \tilde{\beta}_t \mathbf{I}), \\
p_\theta(x_{t-1}|x_t) &= N(x_{t-1}; \mu_\theta(x_t, t), \textstyle\sum_\theta(x_t, t)),
\end{aligned}
\tag{7}
$$

where $\tilde{\mu}_t(x_t, x_0) := \frac{\sqrt{\alpha_{t-1}}\beta_t}{1-\alpha_t}x_0 + \frac{\sqrt{1-\beta_t}(1-\alpha_{t-1})}{1-\alpha_t}x_t$, $\tilde{\beta}_t := \frac{1-\alpha_{t-1}}{1-\alpha_t}\beta_t$, and the variance $\sum_\theta(x_t, t) = \sigma_t^2 \mathbf{I}$. Ho et al. (Ho et al., 2020) set $\sigma_t^2 = \beta_t$. Thus, we can rewrite $L_t$ as follows:

$$
\begin{aligned}
L_t &= \mathbb{E}_{x_0,\epsilon}\left[\frac{1}{2\sigma_t^2}\left\|\frac{\sqrt{\alpha_{t-1}}\beta_t}{1-\alpha_t}x_0 + \frac{\sqrt{1-\beta_t}(1-\alpha_{t-1})}{1-\alpha_t}x_t - \boldsymbol{\mu}_\theta(\mathbf{x}_t, t)\right\|^2\right] + C \\
&= \mathbb{E}_{x_0,\epsilon}\left[\frac{\beta_t^2}{(1-\beta_t)(1-\alpha_t)}||\epsilon - \epsilon_\theta(x_t, t)||^2\right] + C.
\end{aligned}
\tag{8}
$$

The denoising network is indeed optimized to approach $x_0$, and $\epsilon$ can also be employed as training target with a deterministic relationship to $x_0$. Ho et al. (Ho et al., 2020) demonstrated that using $\epsilon$ as the training target empirically outperforms training directly to predict $x_0$. Additionally, they empirically observed that the simplified objective (Eq. 1) with constant-weighting form yields better sample quality, which subsequently becomes the default training objective of diffusion models.

## 3.2 CONSTANT WEIGHTING INTRODUCES BIAS IN TRAINING

We treat $\epsilon$ as the explicit and direct target, and $x_0$ as the implicit but intrinsic target. Given the predicted noise $\epsilon_\theta(x_t, t)$ of the denoising network, we can derive $\hat{x}_0$ from Eq. 3 as follows:

$$
\begin{aligned}
\hat{x}_0 &= \frac{1}{\sqrt{\alpha_t}}x_t - \frac{\sqrt{1-\alpha_t}}{\sqrt{\alpha_t}}\epsilon_\theta(x_t, t) \\
&= \frac{1}{\sqrt{\alpha_t}}(\sqrt{\alpha_t}x_0 + \sqrt{1-\alpha_t}\epsilon) - \frac{\sqrt{1-\alpha_t}}{\sqrt{\alpha_t}}\epsilon_\theta(x_t, t) \\
&= x_0 + \frac{\sqrt{1-\alpha_t}}{\sqrt{\alpha_t}}(\epsilon - \epsilon_\theta(x_t, t)) \\
&= x_0 + \frac{1}{\sqrt{SNR(t)}}(\epsilon - \epsilon_\theta(x_t, t)).
\end{aligned}
\tag{9}
$$

Further, we can rewrite Eq. 9 to express $x_0$ in terms of two components: the *estimated $\hat{x}_0$* part and the *amplified error* part.

$$x_0 = \underbrace{\hat{x}_0}_{estimated\ \hat{x}_0} + \underbrace{\frac{1}{\sqrt{SNR(t)}}(\epsilon_\theta(x_t, t) - \epsilon)}_{amplified\ error}. \tag{10}$$

Although the difference between the predicted $\epsilon_\theta(x_t, t)$ and the target Gaussian noise $\epsilon$ may be very small at every step, the amplification coefficient $\frac{1}{\sqrt{\text{SNR}(t)}}$ is expected to be significantly larger as the step $t$ increases (as shown in Fig. 1), which would result in a substantial deviation of the estimated $\hat{x}_0$ from the target $x_0$. We also visualize the estimated $\hat{x}_0$ and the amplified error at different timesteps via feeding $x_t = \sqrt{\alpha_t}x_0 + \sqrt{1-\alpha_t}\epsilon$ into the denoising network once. The *estimated $\hat{x}_0$* increasingly deviates from the ground-truth $x_0$ when $t$ grows larger, meanwhile the *amplified error* becomes larger and even gradually approaches $x_0$. In this regard, we can find that the constant training weight strategy is indeed biased, and optimizing the explicit target $\epsilon$ uniformly across different timesteps cannot guarantee approaching the implicit target $x_0$ in an optimal manner.

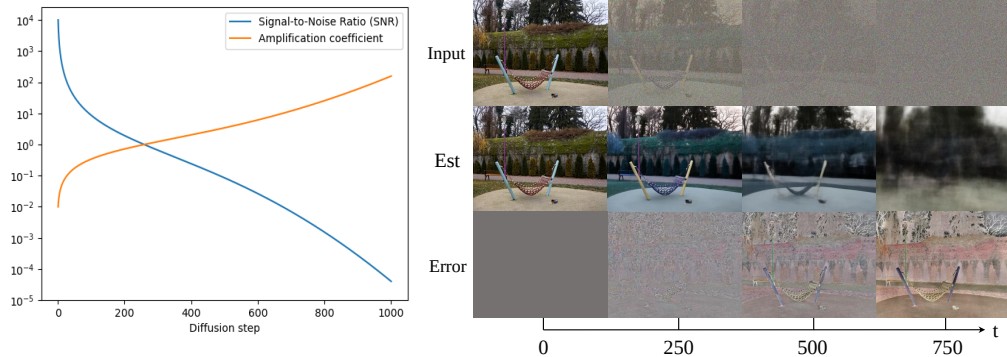

Figure 1: Left: The visualization of SNR($t$) and amplification coefficient $\frac{1}{\sqrt{\text{SNR}(t)}}$ at different timesteps. Right: The upper row is the input $x_t$ at different timesteps. We employ the diffusion model (Dhariwal & Nichol, 2021) pretrained on ImageNet dataset to obtain the *estimated $\hat{x}_0$* part and *amplified error* part of each input $x_t$. The second row is the *estimated $\hat{x}_0$*. The bottom row is the corresponding *amplified error* part. Apparently, as step $t$ gets larger, the *estimated $\hat{x}_0$* severely deviates from $x_0$ and the *amplified error* part gradually approaches $x_0$.

### 3.3 DEBIASED TRAINING STRATEGY

The above theoretical analysis provides a principled guidance to eliminate the biased estimation problem. Concretely, we not only expect the loss function to reach the explicit target $\epsilon$ but also encourage the estimated $\hat{x}_0$ to approach the implicit target $x_0$. Therefore, it is essential to take into account the varying impact of noise prediction at different steps $t$ when designing the loss function. In this regard, we propose a debiased loss formulation which is simple but effective by taking the amplified coefficient into account:

$$L = \sum_t \mathbb{E}_{x_0, \epsilon} \left[ \frac{1}{\sqrt{SNR(t)}} ||\epsilon - \epsilon_\theta(x_t, t)||^2 \right]. \tag{11}$$

In other word, we assign higher weight as the step $t$ increases (i.e., when adding more noise to $x_0$), thereby compelling the noise error $(\epsilon_\theta(x_t, t) - \epsilon)$ to decrease more significantly at larger step $t$. Note that we use the above form instead of the seemingly more reasonable weighting $\frac{1}{SNR}$ as it would hinder the optimization of the explicit target $\epsilon$. We explain it in detail in Appendix A.

## 4 COMPREHENSIVE UNDERSTANDING THE BIAS PROBLEM

In this section, We aim to address several key questions crucial for achieving a systematical understanding of the bias problem in conventional diffusion models: Why is the bias problem important? What are its effects? And what is the underlying cause? We believe answering these questions is essential for unraveling and dissecting the black box of diffusion models.

### 4.1 BIASED ESTIMATION IN THE TRAINING PROCESS

First, we illustrate the one-step estimation $\hat{x}_0$ in Fig. 2 to compare the results obtained using the original constant weighting and our variant. There is a general tendency for the estimated $\hat{x}_0$ of both weighting strategies to gradually deviate from the original $x_0$ as the step $t$ increases, which is inevitable due to the increasing noise in the input $x_t$. However, when utilizing the constant weighting loss for training, noticeable color shifts and inferior arrangement of human faces can be observed in the early steps ($t = 999$ and $t = 950$), severely deviating from the target $x_0$. In contrast, our strategy effectively reduces the bias, achieving greater consistency with the targets across various timesteps, even under relatively high noise levels (e.g., at $t = 999$ and $t = 900$). These findings indicate that the proposed debiased formulation facilitates training in a more appropriate direction. More analyses are available in Appendix B.

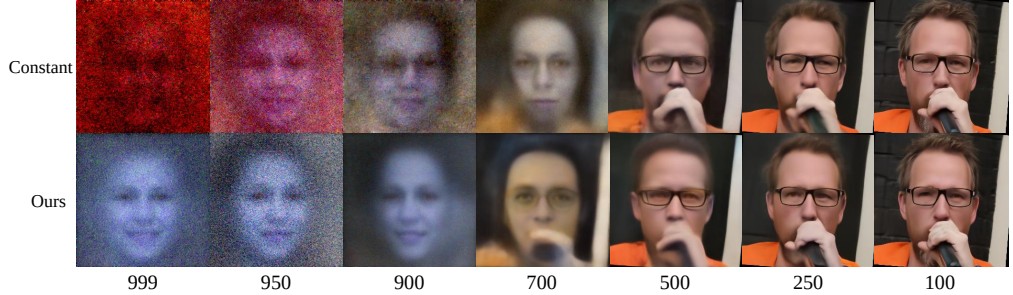

Figure 2: We present the one-step estimation results of $\hat{x}_0$ using different input samples $x_t$, where the diffusion models are pretrained on the FFHQ dataset (Karras et al., 2019) with different loss weighting strategies. One-step estimation: start from a clean image and add noise to get $x_t$ according to Eq. 3. Then put $x_t$ into the denoising network once to get the estimated noise, and finally get the estimated $\hat{x}_0$ by reversing Eq. 3. The top row displays the results obtained using a well-trained constant weighting model, while the bottom row showcases the results achieved with our well-trained debiased weighting model.

## 4.2 BIASED GENERATION ON THE SAMPLING PROCESS

We further analyse the detrimental effects of the biased estimation problem introduced by the constant weighting loss for model inference, i.e., *biased generation* on the sampling process. As seen in the first two rows in Fig. 3, biased generation primarily attributes to the chaos and inconsistency in the early few sampling steps, which substantially affects the final generation through error propagation. We particularly observe pronounced color shifting in biased generation when employing a small number of sampling steps (e.g., $T = 2$), which remains challenging to correct even with an extended sampling process (e.g., $T = 1000$). In contrast, training with our strategy can essentially prevent the issue (e.g., the shown images with $T = 2$), eliminating the need for a lengthy correction process. Moreover, generated images using our strategy show enhanced details and global consistency compared to the baseline method. More analyses are available in Appendix C.

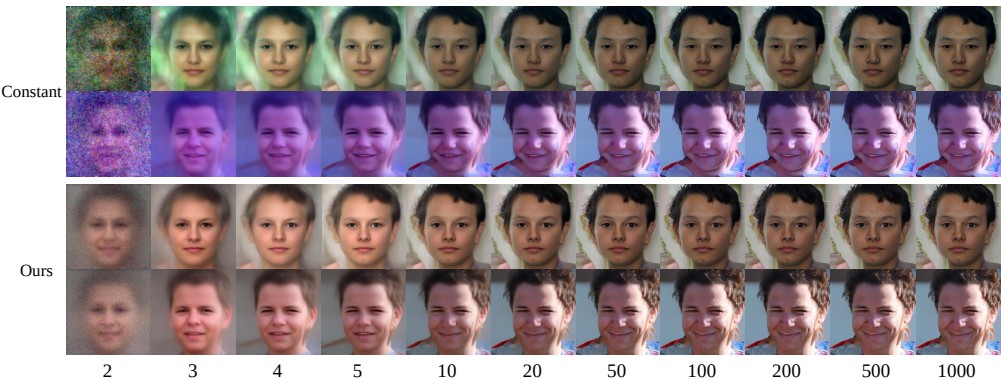

Figure 3: Sampling results with different total steps $T$. The upper two rows are the generated samples of constant weight. The bottom two rows are the generated samples of our improved version.

## 4.3 UNRAVELING THE UNDERLYING CAUSES OF BIASED ESTIMATION

Finally, we take one step further to unravel the underlying causes of biased estimation. Specifically, the optimization difficulty and importance of the denoising network is vastly different across step $t$.

**Different optimization difficulty** Intuitively, the input $x_t$ is closer to the target as step $t$ becomes larger. Consequently, the network encounters varying levels of fitting difficulty across different values of $t$, with larger values of $t$ being relatively easier. To verify this, we plot the Mean squared error

(MSE)-step curve under several settings in Fig. 4. In the "Initial" setting, the MSE value is directly computed between the network input $x_t$ and the target Gaussian noise. The remaining two settings compute the MSE value between the network output and the target Gaussian noise, with "Constant" representing the constant weighting method and "Ours" representing the debiased weighting strategy. The distribution of MSE value under "Initial" mode is extremely unbalanced, in which the MSE value is negligible when $t > 600$. Consequently, this imbalance endows different optimization difficulty across step $t$ and renders the constant weighting strategy suboptimal. Specifically, when $t$ is sufficiently large, the MSE value between the network input and the target becomes extremely small, allowing the network to "do nothing" while still maintaining a low MSE loss.

The above analysis is verified in Fig. 4. For $t > 950$, the MSE value in constant weight mode surpasses that of the "Initial" mode, indicating the output deviates even further from the target than the input. This observation illustrates that the denoising network in constant weight setting fails to identify the noise pattern in the input and, therefore, cannot effectively handle the denoising task. In contrast, our weight strategy consistently yields MSE values lower than those of the "Initial" mode, demonstrating its exceptional denoising capability, particularly for highly noisy inputs.

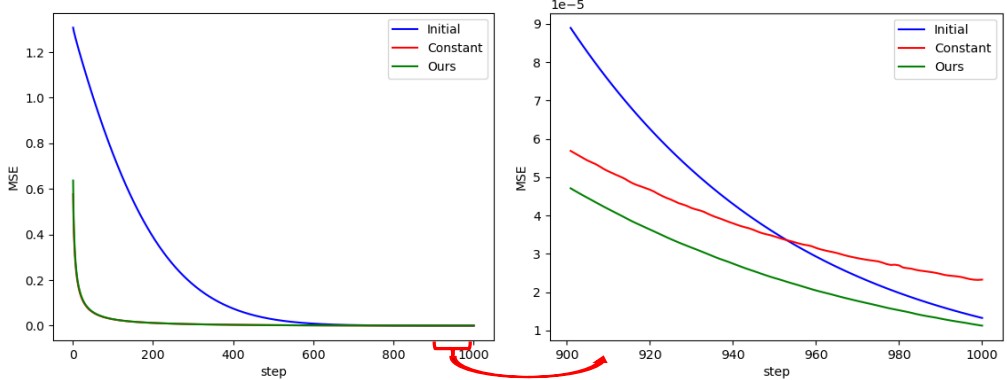

Figure 4: MSE-step curve under several settings. "Initial" mode is calculated between input and target. Obviously, the optimization difficulty is vastly different across step $t$. "Constant" and "Ours" modes are calculated between network output and target. "Constant" denotes constant weight strategy and "Ours" stands for our debiased weight strategy. Note that "Constant" and "Ours" visually overlap in the left figure due to large scale .

**Different importance** The importance of the denoising network varies across step $t$. Intuitively, initial steps are important for both training and sampling process. For training, the initial steps pose greater difficulty due to the presence of high noise levels in the input. For sampling, the initial steps serve as the foundation for subsequent steps, contributing to error propagation. Theoretically, we have verified that initial steps should be emphasized to reach the implicit target $x_0$ in section 3. Additionally, we also find evidence supporting the crucial role of initial steps in diffusion models (Nichol & Dhariwal, 2021; Wang et al., 2023). For example, Nichol et al. (Nichol & Dhariwal, 2021) demonstrated that the first few steps of the diffusion process contribute the most to the variational lower bound. Wang et al. (Wang et al., 2023) found that reusing update directions from initial steps with adaptive momentum sampler can generate images with enhanced low-level details. The constant weighting strategy assumes equal importance across all steps. While, our method assigns higher weights to the initial steps, which is consistent with both intuition and theory.

## 5 EXPERIMENTS

### 5.1 SETUP

**Datasets.** We perform experiments on unconditional image generation using the FFHQ (Karras et al., 2019), AFHQ-dog (Choi et al., 2020), and MetFaces (Karras et al., 2020a) datasets. These datasets contain approximately 70k, 50k, and 1k images respectively. We resize and center-crop data to 256×256, following the pre-processing performed by ADM (Dhariwal & Nichol, 2021).

**Training Details.** We set T = 1000 for all experiments. We implement the proposed approach on top of ADM (Dhariwal & Nichol, 2021), which offers well-designed architecture and efficient sampling. We train our model for 500K iterations with a batch size of 8.

**Evaluation Settings.** Following the common practice (Song & Ermon, 2020), we utilize an Exponential Moving Average (EMA) model with a rate of 0.9999 for all experiments. Besides, we generate 50K samples for each trained model and use the full training set to compute the reference distribution statistics, following (Ho et al., 2020; Choi et al., 2022). During inference, we obtain results with fewer sampling steps than T by employing the respacing technique. For quantitative evaluations, we employ the Fréchet Inception Distance (FID) (Heusel et al., 2017).

## 5.2 COMPARISON TO EXISTING WEIGHTING STRATEGIES

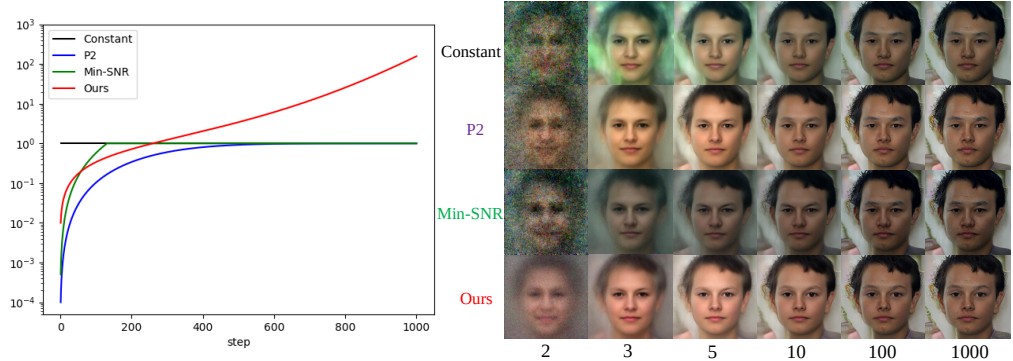

Figure 5: Left: Visualization of various weighting strategies. P2 and Min-SNR starts from the basis of constant weight and lower the weight down for small $t$. Right: Sampling results with different total sampling steps $T$. From top to bottom, they are constant, P2, Min-SNR, and our method. Evidently, P2 and Min-SNR still suffer from bias and artifacts during the initial generation stage.

**Unified perspective on existing weighting strategies** The majority of existing methods (Ho et al., 2020; Dhariwal & Nichol, 2021; Rombach et al., 2022) adhere to the prevailing training objective of predicting Gaussian noise using a constant weight. Several prior methods (Choi et al., 2022; Kingma et al., 2021; Salimans & Ho, 2022; Hang et al., 2023) investigate alternative training targets and weighting strategies. For example, P2 (Choi et al., 2022) proposes a weighting scheme to prioritize higher noise levels that recover content information. Min-SNR (Hang et al., 2023) treats the diffusion training as a multi-task learning problem and designs a weighting strategy to avoid the model focusing too much on small noise levels. Salimans et al. (Salimans & Ho, 2022) proposed some more weighting strategies. We present a visualization of these distinct weighting strategies in Fig. 5 and refer to them as SNR-aware weighting strategies.

Our method differs from these existing methods in several key aspects. (1) Most importantly, they struggle to uncover the core: existing constant weighting strategy is biased and sub-optimal. Thus, they lack the ability to provide insights and future directions to the followers.(2) They intuitively modified the weight on the basis of the sub-optimal constant weighting. For instance, compared with constant weight, they only lower the weight for small $t$, keeping the weight unchanged for the remaining substantial portion of the steps. Consequently, they encounter difficulties in establishing general principles for guiding the design of the weighting strategy.

A unified perspective can be adopted on these existing SNR-aware weighting strategies within the framework of the debiased principle. Our theoretically unbiased principle elucidates that the weight should monotonically increase and assign higher weights to large $t$, similar to the red curve depicted in Figure 5. And prior methods tend to assign lower weights to small $t$ values, adhering to the principle overall. This also elucidates the underlying nature and rationale for their superior performance compared to constant weight. However, without identifying the inherent biased problems, these intuitively designed weighting methods still cannot achieve optimal performance. We also compare different training targets in Appendix D.

**Quantitative comparison.** Tab. 1 presents a quantitative performance comparison of various weighting strategies across different sampling steps $T$. Our method achieves the highest performance across multiple datasets and sampling steps. It is worth noting that the performance gain is particularly pronounced with smaller datasets and shorter sampling steps. This observation indicates the generalization and robustness of our method.

Table 1: Quantitative comparison. The experimental results are reported in terms of FID under a fair setting, with the only distinction being the loss weighting strategy. * denotes the results reported in the original paper. However, as certain essential training details of P2* (e.g., training iterations) are unknown, its reported values are used for reference only.

| Dataset | Step $T$ | Constant | P2 | P2* | Min-SNR | Ours |
|---------|----------|----------|-----|-----|---------|------|
| FFHQ | 1000 | 10.8636 | 6.5173 | 6.92 | 6.5012 | 6.3537 |
| | 500 | 11.0266 | 6.7919 | 6.97 | 6.8733 | 6.7055 |
| | 250 | 11.7802 | 7.4777 | - | 7.7219 | 7.3854 |
| | 100 | 15.6705 | 10.8546 | - | 11.3910 | 10.8154 |
| | 50 | 22.3752 | 16.5376 | - | 17.3284 | 15.3447 |
| | 20 | 41.2703 | 34.3992 | - | 34.6515 | 29.3796 |
| AFHQ-dog | 1000 | 18.2999 | 17.0680 | 11.55 | 17.3418 | 14.9284 |
| | 500 | 18.6062 | 17.4743 | - | 17.6393 | 14.9461 |
| | 250 | 19.1036 | 17.7591 | 11.66 | 17.9216 | 15.0327 |
| | 100 | 20.4464 | 18.3439 | - | 18.4205 | 15.8210 |
| MetFaces | 1000 | 41.4184 | 14.2044 | - | 30.8761 | 9.1683 |
| | 500 | 42.1150 | 14.4476 | - | 31.1675 | 9.4289 |
| | 250 | 42.3241 | 14.7384 | 36.80 | 31.3396 | 9.8486 |
| | 100 | 42.6236 | 14.9937 | - | 31.6259 | 10.3884 |

**Qualitative comparison.** Fig. 6 presents the qualitative results. As anticipated, the biased constant weighting strategy produces images with inferior global structure and color alignment. P2 and Min-SNR enhance the sample quality by building upon the constant weight foundation. However, they still produce images with inferior global structure. This is due to their significant bias and chaotic behavior during the initial sampling steps, as depicted in Fig. 5.

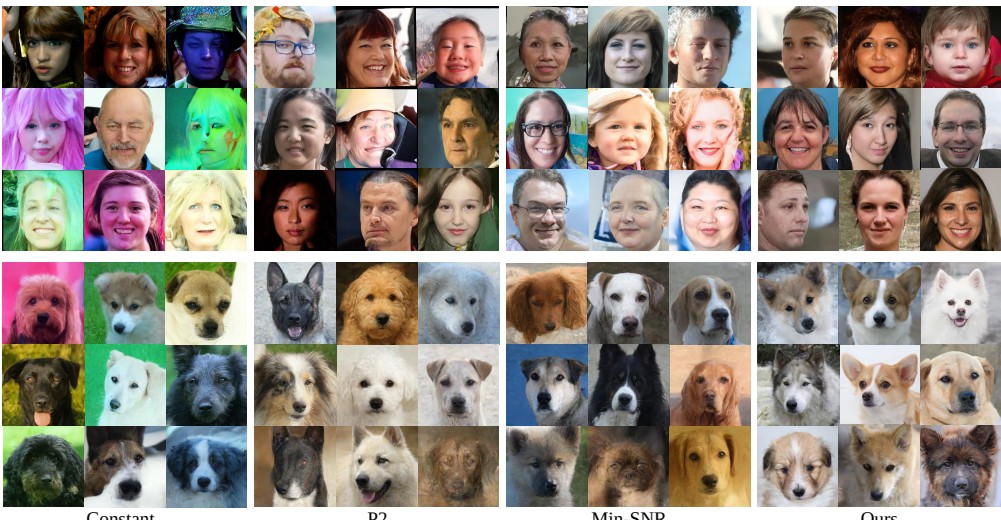

Constant       P2       Min-SNR       Ours

Figure 6: Visual results of different weighting strategies on different datasets. We randomly choose the first nine generated images without cherry-pick. The first row is trained on FFHQ dataset and the second row is on AFHQ-dog dataset.

**High efficiency.** Fig. 7 illustrates the FID-training iterations curve and the FID-sampling steps curve for the FFHQ dataset. The training curve clearly demonstrates the superior efficiency and

potential of our method. For instance, our weighting strategy matches the performance of 1000k iterations of constant weight training with only 400k iterations. In terms of sampling, our method surpasses all existing weight strategies across all sampling steps. Moreover, consistent with the analysis in section 4.2, the performance gains are more pronounced with fewer sampling steps.

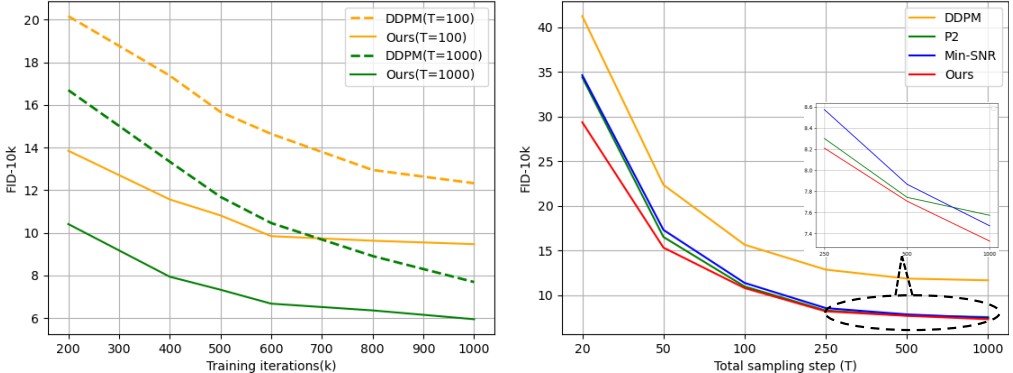

Figure 7: Left: FID-training iterations curve. Right: FID-sampling steps curve. These two curves are obtained on FFHQ dataset. Our method is more efficient and high-performing. Note that, we use DDPM to denote the constant weighting strategy.

## 5.3 COMPARISON TO THE PRIOR LITERATURE

Generative models have enabled unprecedented and photorealistic synthesis (Brock et al., 2018; Karras et al., 2019; Sauer et al., 2021; Esser et al., 2021; Rombach et al., 2022; Ho et al., 2022; Bao et al., 2023). These methods achieve high performance through meticulously crafted architectures and designs. Our method offers a general strategy for diffusion models, achieving competitive performance without relying on intricate techniques. Additionally, our contribution complements these methods and can further extend their performance potential. For example, we achieve substantial improvements by only changing the loss weight on top of ADM. We provide a quantitative comparison and discussion in Appendix E.

## 5.4 BROADER IMPACT AND FUTURE WORK

Given that diffusion models usually serve as fundamental building blocks for various application-oriented works, our method provides valuable inspiration and insights for these endeavors. Additionally, we identify several potential avenues for future research within the community. (1) The elucidated mechanism behind the biased problem offers valuable insights for downstream tasks, such as editing and restoration, facilitating the integration of the bias issue into specific tasks. (2) The biased problem can be investigated from other perspectives, such as noise schedule (Ning et al., 2023; Chen, 2023). It is encouraging to discuss the defects of diffusion models from a unified perspective. For example, one can explore the characteristics and shortcomings from the design space of diffusion models (Karras et al., 2022), and elucidate their relationships and correlations.

## 6 CONCLUSION

This paper provides theoretical analyses and comprehensive studies to demonstrate that the traditional uniform weighting loss function is prone to causing biased estimations during the training of diffusion models, by examining the existence, impact, and underlying reasons behind this issue. To mitigate this problem, we employ a simple yet highly effective weighting strategy that adheres to the theoretically unbiased principle. Empirical studies conducted on multiple datasets, along with comparisons with existing weight methods, further validate the effectiveness of our proposed approach. We also believe these analyses contribute to a deeper understanding of the underlying mechanism of diffusion models.

**Reproducibility Statement**   We are highly confident of the reproducibility of this work. We strictly follow the baseline setting P2 (Choi et al., 2022), with detailed setting description in section 5.1. We employ the official code of P2 and only modify the loss weight with one additional line of code as follows:

```python
# original constant weight
mse_loss = (target - model_output) ** 2

# ours
weight = 1/torch.sqrt(SNR)
mse_loss = weight * (target - model_output) ** 2
```

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

# Appendix

## A  DISCUSSION ON THE THEORETICALLY OPTIMAL WEIGHT: $\frac{1}{SNR}$

As we have mentioned, we didn't employ the seemingly more reasonable loss weight $\frac{1}{SNR}$. We use "SNR" to denote the weight of $\frac{1}{SNR}$. We demonstrate the reason from two aspects. (1) SNR weight damages the explicit target without further boosting the implicit target as shown in Fig. 8. Concretely, the MSE of the SNR mode completely overlaps with our $\frac{1}{\sqrt{SNR}}$ weigh strategy for large $t$, indicating that SNR weight can't further boost the implicit target. However, the MSE of the SNR mode is substantially larger than all other weighting strategies for small $t$, indicating that the explicit target is seriously violated. The reason behind this is the excessive range field of SNR weight ranging from $10^{-4}$ to $10^4$, which causes the denoising network excessively focusing on few early steps. (2) Empirically, the SNR mode performs terribly, as shown in Fig. 9. Min-SNR (Hang et al., 2023) also explores predicting Gaussian noise with the weight of $\frac{1}{SNR}$, and they find that this setting leads to divergence. Thus, our experimental result is also consistent with the conclusion of Min-SNR.

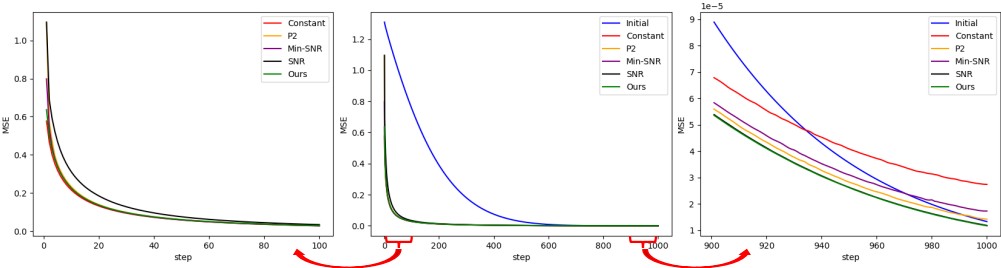

Figure 8: MSE analysis of different weighting strategies. We use "SNR" to denote the weight of $\frac{1}{SNR}$. SNR weight can't further lower the MSE for large $t$. On the contrary, it's denoising ability is worsen for small $t$ with large MSE. The reason behind this is the excessive range field of SNR weight ranging from $10^{-4}$ to $10^4$, which causes the denoising network excessively focusing on few early steps. For example, for batchsize=8, if one $t$ is large and the remaining seven $t$ are small, the network will pay excessive attention to the large $t$ with high weight, while at the cost of sacrificing the seven small $t$s. In contrast, our method achieves the lowest MSE across these weighting strategies at different step $t$, only slightly larger than the constant weight for $t < 50$.

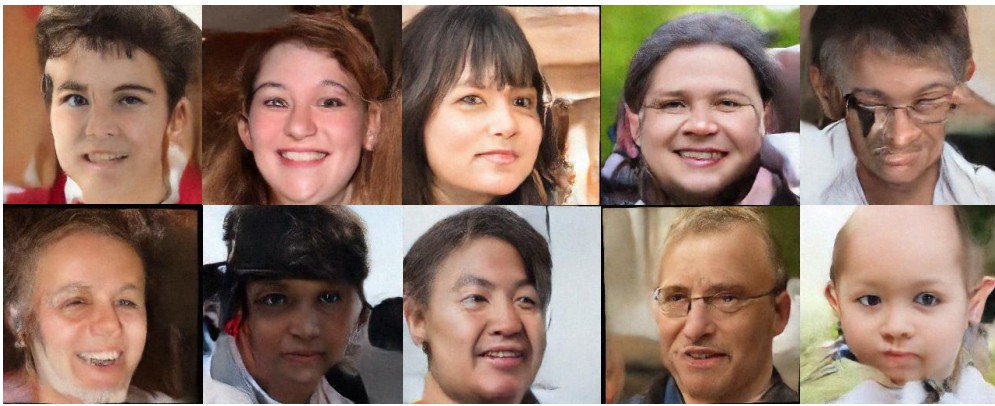

Figure 9: First ten generated sample of "SNR" weight. This weight strategy leads to divergence and poor sample quality.

## B MORE ANALYSES OF THE BIASED ESTIMATION IN THE TRAINING PROCESS

In the main manuscript, we show the existence of biased estimation in the training process with the one-step estimated $\hat{x}_0$. In this section, we further illustrate the bias problem exploiting the intermediate feature maps at different step $t$ in Fig. 10. Fig. 10 is achieved in the same way as Fig. 2 via visualizing the feature maps of the one-step estimation process at various step $t$.

The intermediate feature maps can reflect the structures underlying the noisy samples (Hertz et al., 2022; Tumanyan et al., 2023). Consistent with the conclusion in Fig. 2, the constant weight mode struggles to generate facial structures for large step $t$ ($t > 900$). In contrast, our method can generate clear facial layout even with the most noisy $x_{999}$ as input.

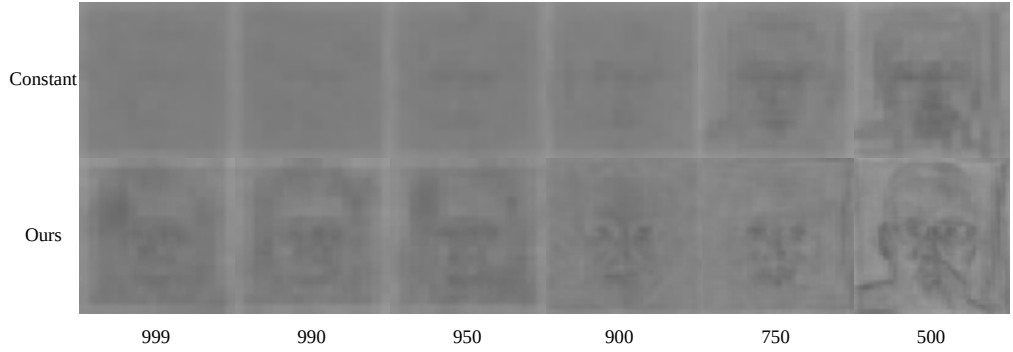

Figure 10: The intermediate feature maps at different steps $t$. Constant strategy struggles to generate clear facial architecture with noisy $x_t$ as input ($t > 900$). In contrast, our method can generate clear facial layout even with the most noisy $x_{999}$ as input.

## C MORE ANALYSES OF THE BIASED GENERATION

In the main manuscript, we indicate the biased generation with generated samples of different total sampling steps $T$. In this section, we show more analyses and visualization of the biased generation. As shown in Fig. 11 and Fig. 12, we respectively show the estimated $\hat{x}_0$ and intermediate feature maps of different weighting strategies at different step $t$. The feature visualization method is similar to prompt-to-prompt (Hertz et al., 2022).

From Fig. 11, we observe that constant weight exhibits artifacts and color shift in the first generation step, resulting in the final generated images with color and structure distortion. P2 and Min-SNR also show global artifacts and inconsistency in the first generation step. Thus, their generated images suffer from poor structures. On the contrary, our method is free of artifacts and color distortion in the whole generation process.

From Fig. 12, we observe that constant, P2, and Min-SNR weight strategies struggle to generate clear facial architecture in the early generation steps. Besides, their intermediate feature maps of the final step also demonstrate poor global consistency. In contrast, our method demonstrates clear facial architecture even at very early steps, and the final feature maps are also more visually pleasing.

## D DIFFERENT TRAINING TARGETS

In this section, we delve into the difference between $x_0$ prediction, $\epsilon$ prediction and $v$ prediction. Most previous works (Nichol & Dhariwal, 2021; Dhariwal & Nichol, 2021; Nichol et al., 2021; Rombach et al., 2022) follow DDPM (Ho et al., 2020) to predict the noise $\epsilon$. Some works (Salimans & Ho, 2022; Gu et al., 2022) use reparameterization to predict $x_0$. And some other works (Salimans & Ho, 2022) employ the network to predict $v \equiv \alpha_t \epsilon - \sigma_t x_0$.

Predicting different targets is mathematically equivalent. However, different prediction targets inherently correspond to different optimizing difficulty. $\epsilon$ prediction is theoretically easiest as the distribution of the optimizing target is simple and fixed. This also explains why predicting Gaussian

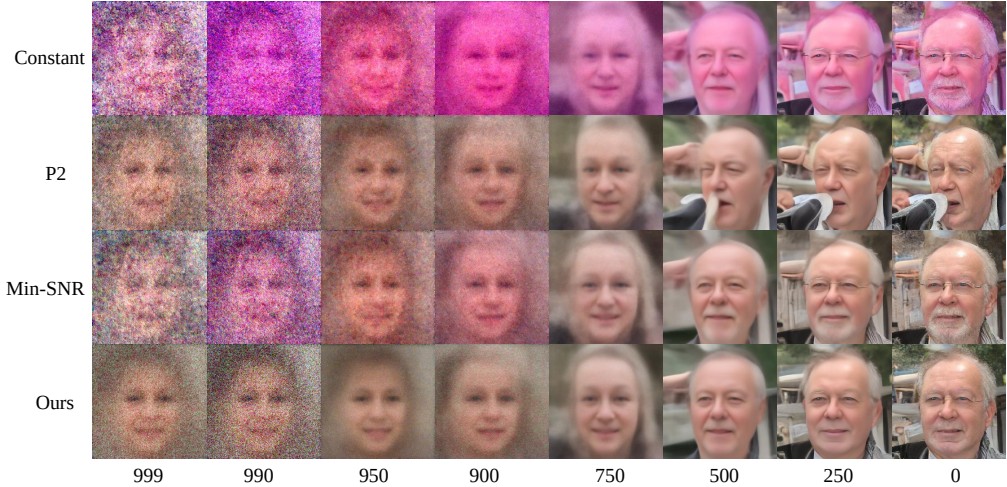

Figure 11: Biased generation: the estimated $\hat{x}_0$ of different weighting strategies at different step $t$. Constant weight exhibits artifacts and color shift in the first generation step, resulting in the final generated images with color and structure distortion. P2 and Min-SNR also show global artifact and inconsistency in the first generation step. Thus, their generated images suffer from poor structures.

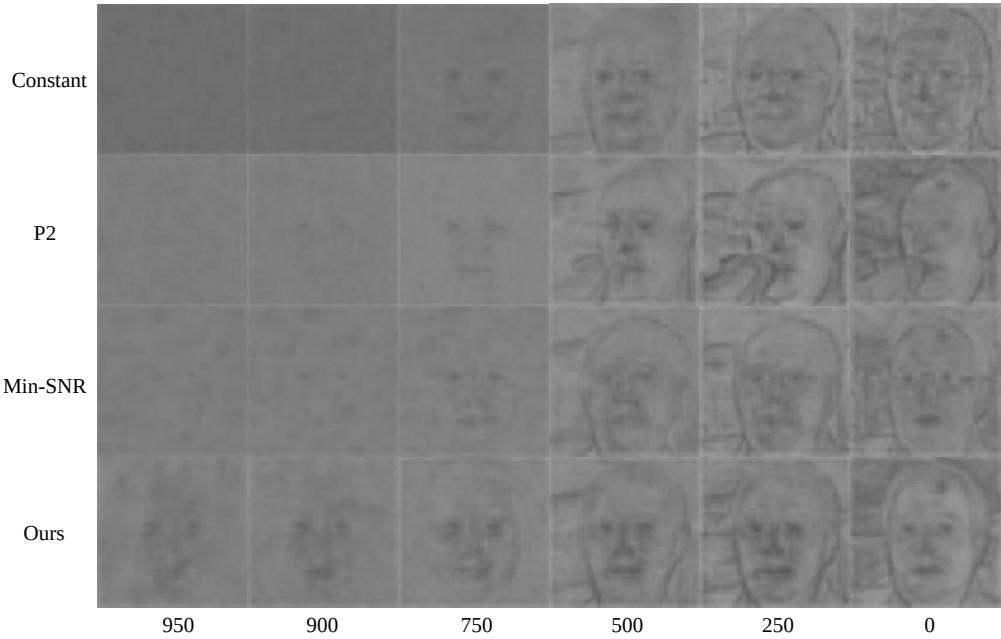

Figure 12: Biased generation: the intermediate feature maps of different weighting strategies at different steps $t$. Constant, P2, and Min-SNR weight strategies struggle to generate clear facial architecture in the early generation steps. Our method shows clear facial architecture even at very early steps.

noise $\epsilon$ with constant weight is most widely employed and becomes the de facto component of diffusion models. Significantly, this further validates the importance and meaning of our work unlocking the biased problem in $\epsilon$ prediction mode.

We also show the performance comparison of different training targets in Tab. 2. Obviously, $\epsilon$ prediction is superior to $x_0$ prediction and $v$ prediction. This is also consistent with the conclusion of (Hang et al., 2023), which finds that predicting $\epsilon$ yields higher quality.

Table 2: Quantitative comparison of different training targets.

|  | $x_0$ prediction | $v$ prediction | $\epsilon$ prediction | Ours ($\epsilon$ prediction with debiased weight) |
|---|---|---|---|---|
| FID | 17.8148 | 19.1426 | 10.8636 | 6.3537 |

## E   COMPARISON TO THE PRIOR LITERATURE

We conducted a comparative analysis between diffusion models trained using our approach and existing models on the FFHQ dataset (Karras et al., 2019), as presented in Tab. 3. Previous generative models(Brock et al., 2018; Karras et al., 2019; Sauer et al., 2021; Esser et al., 2021; Rombach et al., 2022; Ho et al., 2022; Bao et al., 2023) have achieved exceptional results in photorealistic synthesis by employing meticulously designed architectures and methodologies.

In contrast, our method achieves competitive performance with a simple loss weight strategyinstead of relying on intricate techniques. Besides, our method is a general strategy of diffusion models and has the potential to enhance their performance limits. For instance, we achieved substantial improvements by solely adjusting the loss weight on top of ADM (Dhariwal & Nichol, 2021), reducing the FID score from 10.86 to 6.35. Moreover, our method offers the capability to achieve even higher performance. Firstly, we can extend the training duration. For instance, with 500k iterations, our method achieves a FID of 6.35, while with 1000k iterations, it achieves a FID of 4.97. Additionally, we have the flexibility to replace the codebase ADM with a stronger model, such as stable diffusion.

Table 3: Quantitative comparison to prior generative models on FFHQ dataset. Our method is on top of ADM with only one additional line of code, yet achieving substantial performance lift.

| Dataset | Method | Type | FID |
|---|---|---|---|
| FFHQ | BigGAN (Brock et al., 2018) | GAN | 12.4 |
|  | UNet GAN (Schonfeld et al., 2020) | GAN | 10.9 |
|  | StyleGAN (Karras et al., 2019) | GAN | 4.16 |
|  | StyleGAN2 (Karras et al., 2020b) | GAN | 3.73 |
|  | VQGAN (Esser et al., 2021) | GAN+AR | 9.6 |
|  | LDM (Rombach et al., 2022) | Diffusion model | 4.98 |
|  | ADM (Baseline) (Dhariwal & Nichol, 2021) | Diffusion model | 10.86 |
|  | Ours (500k iterations) | Diffusion model | 6.35 |
|  | Ours (1000k iterations) | Diffusion model | 4.97 |

## F   DIFFERENT SAMPLERS

Our weighting strategy is orthogonal to samplers. In this part, we analyse the effect of DDIM sampler (Song et al., 2020) on the sample quality. As shown in Fig. 13, we visualize the generated samples of DDIM sampler under four different weighting strategies. Similar to the conclusion in Fig. 6 of the main manuscript, our method achieves the highest performance with DDIM sampler among these four weighting strategies.

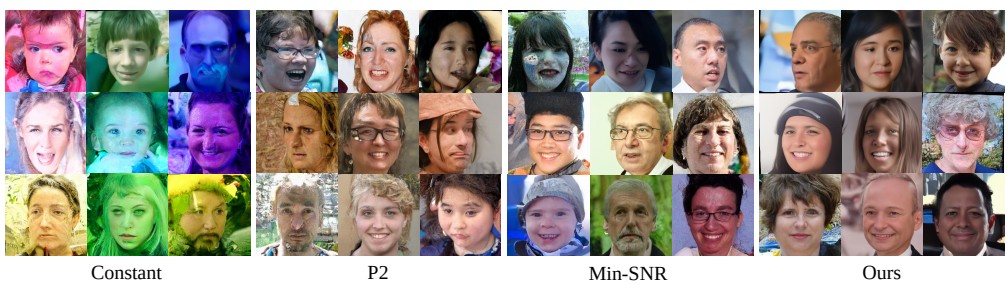

| Constant | P2 | Min-SNR | Ours |

Figure 13: The generated samples of DDIM sampler under four different weighting strategies.

# G    MORE VISUAL RESULTS

In this part, we show more visual results of different weighting strategies on various datasets to further validate the effectiveness and robustness of our method. Fig. 14, 15 , and 16 show the visual results on FFHQ (Karras et al., 2019), AFHQ-dog (Choi et al., 2020), and MetFaces (Karras et al., 2020a) datasets, respectively.

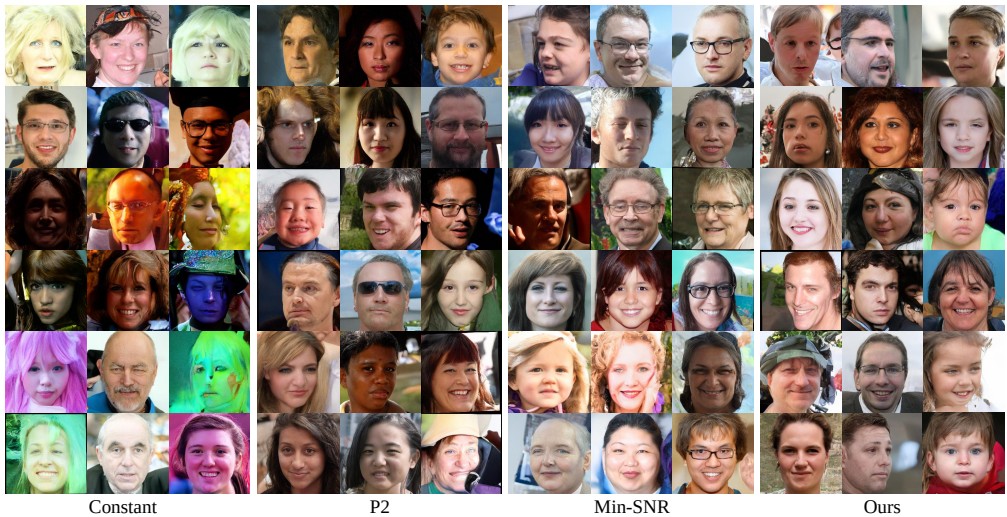

Figure 14: More visual results on FFHQ dataset.

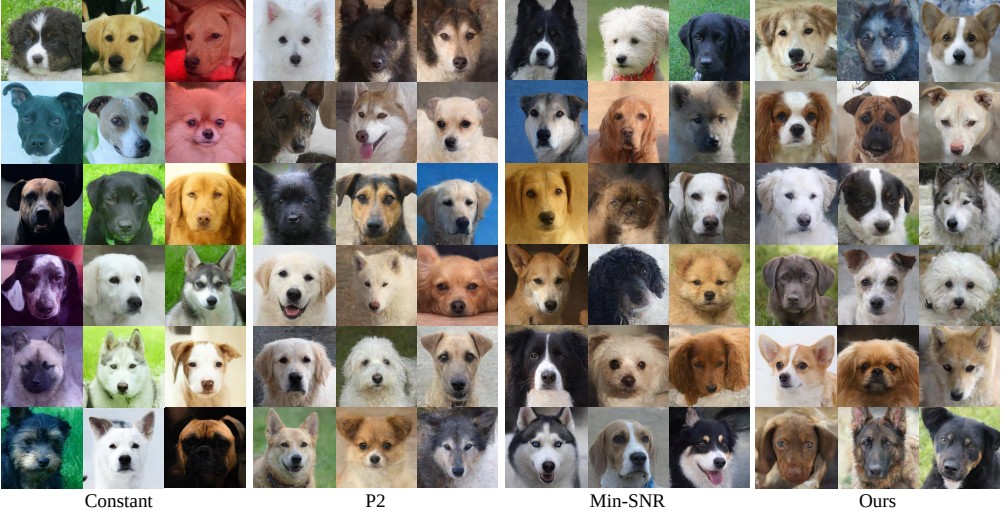

Figure 15: More visual results on AFHQ-dog dataset.

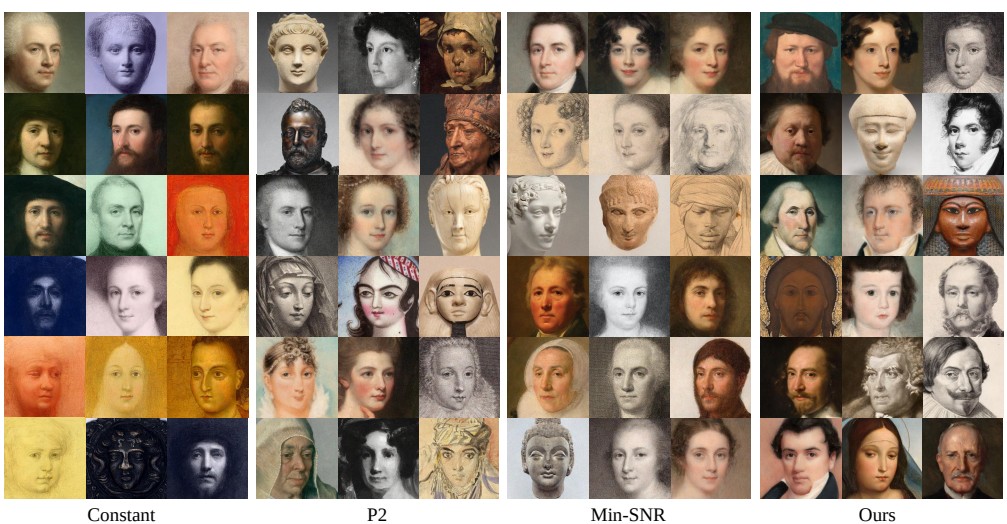

Constant        P2        Min-SNR        Ours

Figure 16: More visual results on MetFaces dataset.

