# OpenReview forum: "Debias the Training of Diffusion Models"
_ICLR.cc/2024/Conference — ICLR 2024 Conference Withdrawn Submission_

### Official Review · Reviewer_kayo · 2023-10-27

**Soundness:** 3 good
**Presentation:** 3 good
**Contribution:** 2 fair
**Rating:** 5
**Confidence:** 4

**Summary:**

This paper demonstrates that the use of a constant loss weight strategy in traditional diffusion models leads to biased estimation during the training phase. To remedy this, this paper proposes a weighting strategy grounded in the theoretically unbiased principle to address this problem. Furthermore, it conducts a thorough and systematic investigation to analyze the inherent bias issue resulting from constant weight loss from multiple perspectives. Finally, the effectiveness of the method proposed in this paper is confirmed through experiments.

**Strengths:**

1.This paper exhibits a concise and lucid narrative style in presenting the methodology, rendering the proof process accessible to a wide readership.

2.The proposed methodology unequivocally enhances the performance of diffusion models with much less training iterations and sampling steps.

3.This paper offers a comprehensive exposition of the bias issue within traditional diffusion models, thereby facilitating a deeper comprehension of diffusion models.

**Weaknesses:**

1.This study exhibits a deficiency in the comprehensiveness of its experiments, and it lacks validation on commonly used benchmark datasets, such as CIFAR-10 and ImageNet. Is this attributed to limitations in the scalability of the proposed methodology? I suggest to conduct more experiments to prove the scalability of the proposed method.

2.In equation (8), the coefficient $\frac{1}{2\sigma ^{2}}$ has been omitted. Though it has no effect on the overall proof, it is better to present this one.

3.This paper, while providing an exposition on biased estimations in diffusion models, places greater emphasis on comparative illustrations with experimental results, thereby rendering the theoretical evidence somewhat less robust.

4.How is the stability of the trainging process? Can you achieve consistent results in every experiment with identical initial conditions?

5.What is the difference of the weighting schedule campare to [1]? It seems the two are very similar and [1] consider more complex situation.

[1] J. Choi, J. Lee, C. Shin, S. Kim, H. Kim, and S. Yoon. Perception prioritized training of diffusion models. In Proceedings of the IEEE/CVF Conference on Computer Vision and Pattern Recognition, pages 11472–11481, 2022.

**Questions:**

The same as Weaknesses.

---

> ### Author Response · Authors · 2023-11-13
>
> Thanks for acknowledging the meaning of our work as well as your careful and constructive suggestions.
>
> 1.	Experiments. We follow the experimental setting and datasets selection of the baseline work P2 [1]. We also conduct more experiments on CelebA-HQ [2] dataset after submission, and the results are shown in the following table.
> |  Steps T  | Constant  | P2  | Min-SNR  | Ours  |
> |  :------  | :------: |  :------:  | :------:  |  :------: |
> | 1000  |  9.3739  | 7.2575   | 6.3218  | 5.9795 |
> | 500   |  10.2355 | 7.7183   | 6.9225  | 6.5717 |
> | 250   |  11.0974 | 8.4334   | 8.0162  | 7.6038 |
> | 100   |  12.0064 | 9.2972   | 9.3851  | 8.8362 |
>
>     The employed FFHQ, CelebA-HQ, and AFHQ are all common, representative and high-quality benchmark datasets. The CIFAR-10 dataset is somewhat outdated. While, we will also complete the experiments on these two datasets as you suggested. While, need to emphasize that existing experimental datasets are already diverse and representative, as well as following the common practice of baseline works [1].
>
>      Besides, there is no need to worry about the reproducibility and scalability of this work. The derivation and analyses are general for constant weighting strategy in diffusion models. We do not impose any assumptions and constrains on the generality and scalability.
>     Additionally, our method only modifies the training weight strategy of the ADM codebase, and thus is of high reproducibility. We will also soon release the code and pretrained models.
>
> 2.	The combination of theory, analyses and experimental results. This paper is not a purely theoretical work. Indeed, the derivation in the beginning is only the start of this work. We hope to provide researchers with comprehensive and in-depth insight and understanding on the bias problem. Thus, we conduct substantial analyses and experiments. Besides, the consistency between theory and experiments is also more convincing.
>
> 3.	Stability of the training. Yes, we achieve consistent and significant performance improvement on all the performed experiments in this paper, regardless of the datasets, total sampling steps, and samplers. These are all clearly shown in the experimental part and the supplementary material.
> Besides, there is no need to worry about the reproducibility of this work. Our method only modifies the training weight strategy of the ADM codebase, and thus is of high reproducibility. We will also soon release the code and pretrained models.
>
> 4.	Comparison with baseline work [1]. In this paper, we move the comparison with related works into experimental part section 5.2. This may cause the confusion of you and the other reviewers.
> Actually, we employ [1] as our baseline. The main difference between [1] and our work is that [1] designed their weighting strategy intuitively and fail to give the insight and analyses of the bias problem in the previous constant weighting strategy.
>
> [1] Choi J, Lee J, Shin C, et al. Perception prioritized training of diffusion models. CVPR. 2022.
>
> [2] Karras T, Aila T, Laine S, et al. Progressive growing of gans for improved quality, stability, and variation. ICLR. 2018.

---

### Official Review · Reviewer_dmLU · 2023-10-29

**Soundness:** 4 excellent
**Presentation:** 4 excellent
**Contribution:** 4 excellent
**Rating:** 8
**Confidence:** 5

**Summary:**

This paper discusses the training weighting of the loss of diffusion models. They theoretically find that original constant weighting strategy is suboptimal, and further propose an improved training loss weight strategy. Besides, they give in-depth analyses of the sub-optimality of the constant weighting strategy from the perspective of existence, impact and reasons. The effectiveness of the proposed method is verified on several datasets.

**Strengths:**

1. It is an interesting idea to analyze the sub-optimality of the training loss weight. This paper theoretically reveal the inherent bias of constant weighting strategy, and propose a debiased principle on the design of the training weight.
2. The in-depth analyses of the sub-optimality of the constant weighting strategy from the perspective of existence, impact and reasons are impressive and insightful. These analyses provide valuable insights and inspirations on the opague generation process of diffusion model.
3. The experiments are solid. The proposed method gains substantial performance improvement on several datasets via simply modifying the training loss weight. The reproducibility is well guaranteed.
4. This paper is well-organized and easy to read and understand.

**Weaknesses:**

1. Allocating higher weight to large t will improve the overall performance. While, what is effect of allocating lower weight at small t. It seems that the MSE error is slightly higher at t=0 than the constant weighting strategy in fig. 4.
2. The visual difference between different baselines and the proposed method seems not obvious in fig. 5.
3. Minor suggestions. The “DDPM” is also used to denote constant weight in fig. 7. The authors are advised to general denotation to avoid confusion.

**Questions:**

I am curious about the performance of treating the original image as training target and its comparison with noise-prediction mode.

---

> ### Author Response · Authors · 2023-11-13
>
> We appreciate your responsible reviewing and the insightful comments.
>
> 1.	Yes, as you observed, allocating lower weight for small t will slightly influence the optimization of the network at small steps. While, the optimization difficulty and importance of the denoising network are inherently different across step t, with small step less important and relatively redundant. This is also consistent with the experiments, where our weighting strategy achieves highest performance.
> 2.	The example in Fig. 5 is randomly selected, while we can still see obvious difference, especially in the small steps. The randomness but superiority indeed effectively validate the generality and robustness of our method.
> 3.	The comparison between different training targets is shown in the supplementary material section D. Our method significantly elevates the performance of the base $\epsilon$-prediction mode, and also largely surpasses the performance of other training targets.

---

### Official Review · Reviewer_kAkY · 2023-10-31

**Soundness:** 2 fair
**Presentation:** 2 fair
**Contribution:** 2 fair
**Rating:** 3
**Confidence:** 4

**Summary:**

This submission deals with the bias of using constant weights for the denoising loss of diffusion models in the training phase …it first identifies the biased generation issue as a result of constraint weighting that results in artifacts such as poor details, global inconsistency, and color shift. It then proposes a new SNR-based weighting mechanism that lifts the diffusion error to the image space and thus debiases the generation. Experiments with FFHQ, AFHQ-dog, and MetFaces show significant FID gains and sampling steps compared with constant weighting.

**Strengths:**

Improving the training efficiency and sampling quality of diffusion models is a timely topic

Experiments and comparison are extensive and the gains are significant

**Weaknesses:**

The paper misses related work in the literature that have already proposed the idea of SNR-weighting for diffusion models. Glancing through the literature, this reviewer found this related work in [1] that proposes the SNR weighting with the same justifications and derivations for sampling. The major novelties need to be clarified. In particular, the derivations from eq. 9 and 10 are already proposed in [1].

[1] Mardani M, Song J, Kautz J, Vahdat A. A Variational Perspective on Solving Inverse Problems with Diffusion Models. arXiv preprint arXiv:2305.04391. 2023 May 7.

**Questions:**

The abstract is vague and high level. The main idea which is the SNR-based weighting mechanism is not explained well.

Do you use SNR weighting for the sampling phase as well?

---

> ### Author Response · Authors · 2023-11-13
>
> Thanks for your valuable suggestions.
>
> 1.	First, we introduce the differences between the work you mentioned and our method.
> The SNR relationship between $\epsilon$-prediction and $x_0$-prediction is not hard to derive and is not the main contribution of our work. Actually several previous methods get this conclusion [1][2] (including the work you mentioned). The difference is that these works usually stop by this conclusion and apply it to other use cases instead of training diffusion models for image generation. For example, [2] employed this relationship to scale the loss value to the same level between the two loss terms, in the use case of employing pretrained diffusion models as loss function to train the inverse network. Significant differently, this derivation only stands as the beginning of our paper. Our work focuses on the bias problem of constant weighting strategy in the basic and critical $\epsilon$-prediction mode for training the diffusion model from scratch for image generation. The SNR relationship is only employed as weight designing guidance.
> For example, (1) we propose the concept of “estimated $x_0$” part and “amplified error” part and in section 3. (2) We conduct a comprehensive and systematic exploration to dissect the inherent bias problem deriving from constant weighting loss from the perspectives of its existence, impact and reasons in section 4. (3) We validate our design choice with substantial experiments to train diffusion models from scratch in section 5.
> 2.  SNR weighting for the sampling. There may exist misunderstanding of you for the work [2]. The “sampling” process of [2] is indeed  training the inverse network with frozen pretrained diffusion model as loss function network. In our work, we employ SNR weighting as a training strategy for training diffusion model from scratch. Thus, our sampling process is certainly free of training strategy.
> 3.  As you view this related work as the only weakness of our method, we hope our explanation can solve your misunderstanding and confusion.
>
> [1] Rombach R, Blattmann A, Lorenz D, et al. High-Resolution Image Synthesis with Latent Diffusion Models. CVPR. 2022.
>
> [2] Mardani M, Song J, Kautz J, Vahdat A. A Variational Perspective on Solving Inverse Problems with Diffusion Models. arXiv. 2023.

---

### Official Review · Reviewer_sefk · 2023-11-04

**Soundness:** 1 poor
**Presentation:** 2 fair
**Contribution:** 1 poor
**Rating:** 1
**Confidence:** 4

**Summary:**

The paper studies the biased estimation problem of diffusion models, by examining the flaw in the $\epsilon$-prediction. The authors also conduct several empirical analyses to support the bias effect. Empirically, the proposed objective achieves better FID scores across facial datasets.

**Strengths:**

- The paper examines the potential bias problem when using $\epsilon$-prediction.

- Empirically, the proposed weighting scheme outperforms previous ones.

**Weaknesses:**

- **No novelty**: It seems that the paper reinvents a well-established objective in the diffusion models literature -- $x_0$ prediction (see the blog https://medium.com/@zljdanceholic/three-stable-diffusion-training-losses-x0-epsilon-and-v-prediction-126de920eb73). The paper "rediscovered" the relation between $x_0$-prediction and $\epsilon$-prediction. The proposed objective in Eq.11 is actually doing $x_0$-prediction type loss: by setting $\epsilon_\theta = \frac{x_t-\hat{x}_0}{\sigma}$ and $\epsilon=\frac{x_t-x_0}{\sigma}$ one could recover the $x_0$-prediction loss.

One step further, there are already works focusing on combining the strengths of $x_0$-prediction and $\epsilon$-prediction, like the $v$-prediction [1] and the pre-conditioning techniques in EDM [2], in the past year.

- The FID score in Table 1 is way too high in the small NFE regime (NFE<100). It makes the comparison much less convincing.


[1] Progressive Distillation for Fast Sampling of Diffusion Models, Salimans et al.

[2] Elucidating the Design Space of Diffusion-Based Generative Models, Karras et al.

**Questions:**

N/A

---

> ### Author Response · Authors · 2023-11-13
>
> It is unprofessional and irresponsible to leave two related works as the main weakness of this work with only few lines of words, even without carefully reviewing this work. It is even more ridiculous to use entry-level blog to help reviewing and understanding our work.
>
> 1.	We show clear discussions and results comparison with related works in section 5.2 and supplementary material section D. Improvements over the \epsilon-prediction mode can be categorized into two types: weighting strategy and different training targets. We give comprehensive comparison between different weighting strategies in section 5.2. We also present the discussions and experimental results of different training targets in supplementary material, including v-prediction.
> 2.	The recovered relationship between $\epsilon$-prediction and $x_0$-prediction only stands as the beginning part of this paper. While, you treat it as the main contribution of our work with prejudice. Our work focuses on comprehensive analyses and experiments to provide clear insight and understanding of the bias problem concealed within the constant weighting in $\epsilon$-prediction.
> For example, (1) we propose the concept of “estimated x_0” part and “amplified error” part and in section 3. (2) We conduct a comprehensive and systematic exploration to dissect the inherent bias problem deriving from constant weighting loss from the perspectives of its existence, impact and reasons in section 4. (3) We validate our design choice with substantial experiments to train diffusion models from scratch in section 5.

---

> ### Comment · Reviewer_sefk · 2023-11-13
> **Response**
>
> I think the authors have a limited understanding of diffusion models and their techniques (e.g. $x_0$ prediction), which have been extensively discussed and are even covered in entry-level blogs. I recommend that the authors take the time to carefully read such resources to acquire a basic understanding.
>
> As I mentioned in my previous review, **the "new objective" (Eq. 11) proposed, analyzed, and experimented on by the authors is essentially equal to $x_0$ prediction loss, a well-established and widely used technique in practice.** (Indeed, people nowadays use v-prediction and EDM-type preconditions instead of the conventional $x_0$ prediction)
>
> I kindly request that the authors refrain from making unwarranted accusations against the reviewer. I went through your paper carefully, and I believe that the primary issue is quite evident, making it unnecessary for me to point out other specific issues in my previous review. Indeed, it appears that there are several instances that show the authors' lack of basic understanding regarding diffusion models. For example, the difference between $\epsilon$ prediction and $\epsilon$ is not an error but a feature of diffusion models. The optimal $\hat{x}\_0$ should not be $x_0$ itself but rather the mean of the posterior  $E_{p_{0|t}(x_0|x_t)}[x]$. In addition, the experiments are on simple single-mode face datasets, rather than more realistic multi-mode datasets like CIFAR-10, ImageNet, or LAION. The FID score reported in the experiments falls far outside the reasonable range, both for the baseline and the proposed method, making the experimental results unreliable.
>
> Based on the author's response, I have decided to lower my rating. I strongly recommend that the authors take time to acquire relevant knowledge to gain a basic understanding of diffusion models and their associated techniques.